# Noninvasive Analysis of Biological Components Using Simplified Mid-Infrared Photothermal Deflection Spectroscopy

**DOI:** 10.3390/s25144368

**Published:** 2025-07-12

**Authors:** Hiroto Ito, Saiko Kino, Yuji Matsuura

**Affiliations:** Graduate School of Biomedical Engineering, Tohoku University, 6-6-05 Aoba, Sendai 980-8579, Japan; hiroto.ito.r2@dc.tohoku.ac.jp (H.I.); saiko.kino.a3@tohoku.ac.jp (S.K.)

**Keywords:** infrared spectroscopy, noninvasive diagnosis, photothermal deflection spectroscopy, photothermal spectroscopy

## Abstract

We developed a photothermal deflection spectroscopy (PTDS) system for the noninvasive analysis of biological tissue. This system detects heat induced by irradiation with pulse-modulated mid-infrared light as the deflection of a probe laser. The probe light is incident on the sensing element horizontal with respect to its contact surface with the sample. This setup simplifies the optical alignment compared to conventional systems, which require the probe laser to be totally reflected at the prism contact surface and aligned with the point of mid-infrared light irradiation. In this study, we measured the PTDS spectra of biological samples to determine the characteristic features of their infrared absorption. We also compared the measurement reproducibility of two configurations: a horizontal optical path and a total reflection optical path. The horizontal optical path showed greater measurement reproducibility than the total reflection optical path when performing intermittent measurements on the wrist.

## 1. Introduction

There is increasing demand for routine blood component monitoring for the early detection and prevention of lifestyle-related diseases. Biochemical analyzers based on colorimetric reactions are used to analyze various blood components, such as sugar, cholesterol, protein, and enzymes. Following blood collection, the analysis requires complex pretreatment and large equipment. Due to significant time and financial costs, it is difficult to implement a system that can be used on a daily basis.

Spectroscopic analysis is suitable for the structural analysis of molecules constituting a living body [1]. Noninvasive analysis would be feasible if light irradiation from the body surface could be used to analyze the components of subcutaneous interstitial fluid and blood. Compared to Raman spectroscopy [2], infrared spectroscopy provides greater signal intensity, which might facilitate the development of a simple system [3]. Strong, sharp absorption peaks appear in the mid-infrared region (wavelength: 3–12 μm) due to fundamental vibrations of the molecules in various components of living organisms, and thus infrared spectroscopy would be suitable for analyzing the complex mixture of components in blood [4].

Many studies have evaluated analytical methods based on attenuated total reflection (ATR), which involves the combination of a Fourier transform infrared spectrometer (FTIR) and an internal reflection element (IRE) with a high refractive index [5,6]. This combined approach measures the absorption spectrum of a sample according to evanescent light that seeps into the sample when infrared light transmitted in the IRE is totally reflected at the boundary between the sample and the IRE. However, the depth of reach of evanescent light is about 2 μm from the surface of the sample, such that the measurement area is limited to mucous membranes without a keratin layer (thickness: 10–20 μm) for noninvasive analysis. Although our research group found a clear correlation between the ATR spectra of human oral mucosa and blood glucose levels, there were difficulties in exposing the oral mucosa to IRE for reasons of hygiene [7,8,9].

Photothermal spectroscopy is an alternative to ATR for mid-infrared spectroscopy. Photoacoustic spectroscopy (PAS) is a photothermal spectroscopic method in which a microphone detects sound waves generated in the air by the thermal vibration of a sample upon irradiation with low-frequency-modulated mid-infrared laser pulses [10,11]. The depth of penetration into the sample is the light penetration length (generally 20–30 μm), which is dependent on the absorption coefficient of the biological tissue, and this penetration depth is able to reach the subcorneal interstitial fluid [12].

Several studies have applied PAS to noninvasive blood glucose measurement, mostly using quantum cascade lasers emitting in the mid-infrared region as the light source [13,14,15,16]. The PAS system has a relatively simple configuration and can detect signals with high sensitivity, but temperature changes in the photoacoustic cell and water vapor from the sample can be problematic. Pleitez [15] solved the water vapor problem by introducing an open gas cell.

Another method is photothermal deflection spectroscopy (PTDS). This method detects heat generated by the absorption of pump light irradiating a sample, which is converted into a change in the refractive index of the sensing medium in contact with the sample [17,18]. This method enables highly sensitive measurements for very small samples and thin films. Therefore, it is widely used in the field of materials research; e.g., spectroscopic analysis of organic semiconductor materials and amorphous silicon [19,20]. Bauer et al. [21,22] applied this technique to noninvasive blood component analysis and compared the performance of PTDS and PAS. Correlation analysis of invasive blood glucose and measured spectra showed that PTDS had a measurement error comparable to that of PAS. In their system, the near-infrared probe light was totally reflected within a sensing prism in contact with the sample near the point of mid-infrared pump light irradiation to enhance detection sensitivity. However, this method requires perfect alignment of the pump light irradiation point and the probe light reflection point, making optical alignment critical. Additionally, as the detection area where these two points overlap is very small, there are concerns that in-plane inhomogeneity of the target biological tissue will reduce the reproducibility of the measurement.

In this study, we propose a system in which the probe light travels straight near the surface of a sensing element to develop a PTDS system with high measurement reproducibility and simpler alignment configuration. Conventional total reflection PTDS systems require point-to-point alignment, whereas the proposed system requires the matching of only the point (mid-infrared pump light) with the line (probe light). Therefore, it is expected to increase the tolerance of optical alignment and improve measurement reproducibility because a relatively large detection area can be obtained. We present the design of the measurement system and the results of an evaluation of its basic characteristics.

## 2. Materials and Methods

Figure 1 shows a schematic of the proposed PTDS system. The excitation light source was a quasi-continuous wave light from a tunable (wavenumber: 930–1200 cm^−1^) mid-infrared quantum cascade laser (QCL) (Hedgehog; Daylight Solutions, Poway, CA, USA) modulated by an optical chopper. The sample was irradiated from below the heat-sensing element. In conventional solid material evaluation systems using PTDS with visible light or near-infrared light, changes in the refractive index due to temperature increases in the air are utilized. However, in biomedical application systems using mid-infrared light, it is necessary to use a solid heat-sensing medium to securely fix soft samples such as skin. Furthermore, since the thermal optical coefficient dn/dT of common crystals and glasses that transmit mid-infrared light is significantly higher than that of air, it is possible to achieve high sensitivity.

Changes in the refractive index of the element due to heat caused optical path variation of the probe light passing parallel to the element surface. This variation was detected by an optical position detector (PDQ30C; Thorlabs, Newton, NJ, USA), and the photothermal deflection signal was obtained by a lock-in amplifier (LI5645; NF, Yokohama, Japan).

Crystalline ZnS is commonly used as a heat-sensing medium in mid-infrared PTDS. However, in some countries, ZnS crystals are subject to specific handling and disposal regulations due to their environmental impact, which may result in problems with its use in the future. Therefore, we introduced an infrared transparent glass-sensing element (FI-01; Nippon Electric Glass, Otsu-Shi, Japan) with a refractive index of 2.65 (ZnS: 2.20) and a transmission wavelength range of 1–10 µm (ZnS: 0.5–10 µm). This glass is sufficiently stable under normal conditions [23] and does not contain toxic substances (such as arsenic or selenium) commonly used in chalcogenide glass, so there are no issues with biocompatibility. It has high transparency to both the near-infrared light used as a probe light and in the excitation light wavelength band of 8–10 μm. Although this glass material costs about the same as ZnS, it can be molded and processed, making it suitable for mass production in the development of future general-purpose systems. The glass-sensing element was 15 mm in width, 26 mm in length, and 7 mm in height, and the probe light was a semiconductor laser with a wavelength of 1550 nm. For measurement performance comparison, we also used a conventional measurement system in which the near-infrared probe light was totally reflected within a trapezoidal glass prism (Figure 1, inset).

## 3. Results and Discussion

Figure 2 shows the PTDS absorption spectrum of 5% glucose solution divided by that of pure water. The glucose solution was prepared by adding 0.5 g of glucose to 9.5 mL of water. An ATR spectrum of the same solution measured by the FTIR system is shown for comparison. The PTDS spectrum measured using the proposed system showed good agreement with the absorption spectrum measured by the ATR method, confirming that this measurement system can obtain infrared absorption information.

Figure 3 shows a calibration plot created using the absorption peak area at 1035 cm^−1^ from spectra obtained from glucose solutions with different concentrations (0.1–5%). A high degree of linearity between the peak area and concentration was confirmed and the coefficient of determination R^2^ for the regression line was 0.995, and the RMSE, which indicates the accuracy of the regression line, was 0.062. At this stage, the peak was observed in solutions with a glucose concentration of 0.3% or higher. A glucose concentration of 0.3% corresponds to a human blood glucose level of 300 mg/dL, but that sensitivity is still insufficient for detecting fasting blood glucose levels of approximately 80–200 mg/dL in healthy subjects. We are currently studying the optimization of the probe light incident position by simulation calculation to improve the sensitivity.

Figure 4 shows a PTDS spectrum measured on the palm side of the wrist of a healthy adult subject. The measurement was performed using pure water as a reference, and both the measurement site and glass-sensing element were rinsed thoroughly with pure water to remove any sebum contamination before measurement. The absorption spectrum of the same measurement site obtained by the ATR method is shown for comparison. We confirmed that this method provided the same shape as the ATR method for biological samples. The spectral peak at 1030 cm^−1^ was attributed to keratinocytes, at 1078 cm^−1^ to stratum corneum tissue, and at 1120 cm^−1^ to lactic acid. All of these peaks were present in the stratum corneum and on its surface in both spectra. As the depth of light penetration of PTDS into biological tissues is estimated to be 20–30 μm [18], it should be able to detect components in interstitial fluid beneath the stratum corneum. However, the absorption levels of the interstitial fluid beneath the stratum corneum did not appear as distinct peaks in the spectra because they were much lower than that of the stratum corneum, as described above. Therefore, detailed spectral analysis using multivariate analysis is necessary for the study of blood components [24]. We have also applied spectral analysis using partial least squares regression (PLSR) for blood cholesterol analysis [25], and we also utilized machine learning methods such as random forests and neural networks for analysis of infrared absorption spectra of human subjects [26]. In addition, in PTDS targeting multilayered structures, there is the possibility of selecting the measurement depth by changing the modulation frequency [21].

We compared the measurement reproducibility between the horizontal optical path PTDS system proposed in this study and a conventional total reflection optical path system for measuring biological samples. At least four hours after eating, when blood components had stabilized, five measurements were taken on the palm side of the wrist. It took two minutes to obtain one spectrum. Experiments were performed in two ways: continuous measurements, where the spectrum was measured five times in a row while the skin remained in contact with the sensing element; repositioning measurement, where the skin was removed from the sensing element after each spectrum measurement, rinsed with water, and the same site was repositioned.

Figure 5 shows PTDS spectra of the wrist, which was measured five times in continuous measurements. Since baseline fluctuations occur in the measured PTDS spectra, the baseline correction has been performed in the 1005–1152 cm^−1^ range to compensate for this. The baselines are shown as dotted lines in Figure 5 and Figure 6. The spectra have been normalized by the area between the line and the spectra. The results showed almost the same measurement reproducibility for both horizontal and total reflection paths.

Figure 6 shows PTDS spectra of the wrist, measured five times via repositioning measurement. Under these conditions, the horizontal optical path was confirmed to show higher measurement reproducibility.

Table 1 shows the coefficients of variation of peak intensity calculated for the three spectral absorption peaks shown in Figure 5 and Figure 6. In repositioning the wrist, the coefficients of variation of the spectra acquired with the horizontal optical path were lower than those acquired with the total reflection path. The horizontal optical path expanded the effective detection area determined by the overlap of the mid-infrared laser light irradiation point and the optical path of the probe light, such that it suppressed the effects of inhomogeneity and microscopic shape changes of the biological tissue. To further reduce the impact of biological tissue heterogeneity, there is potential to improve measurement accuracy by determining the amount of water and fat in the tissue through tissue modeling based on bioimpedance.

## 4. Conclusions

In this study, we investigated a PTDS system using mid-infrared light with a simpler configuration than used in conventional systems in which the probe light passes horizontally through the sensing element. For the future development of a system with fewer safety concerns, infrared light-transmitting glass was selected as the material for the sensing element. We obtained absorption spectra for an aqueous glucose solution with a concentration of 0.3%. Experiments in human subjects confirmed better reproducibility than that obtained for the conventional system using the total reflection optical path when the target site was changed in each measurement due to the increase in area measured.

The heterogeneity of tissue structure and components has a significant impact on the results when attempting to optically analyze the components of biological tissue. Therefore, it is necessary to suppress the influence of heterogeneity by analyzing a larger area, which generally leads to a reduction in measurement sensitivity due to smaller light fluences. Due to this tradeoff, it is necessary to determine the optimal value for each analysis object and method. Since PTDS does not interfere with electrical measurements, combining it with non-optical measurements such as surface electromyographic (sEMG) makes it possible to detect the effects of tissue structures that cannot be detected by light alone.

The PTDS system proposed in this study has the potential to be applied to wearable sensors for home healthcare, as shown in [27], by miniaturizing the system. To miniaturize the system, we are optimizing the probe light incidence position through simulation calculations and designing an optical fiber slot integrated with a glass-sensing element. This enables the probe light to be incident from an optical fiber equipped with a collimator without alignment, allowing for the development of a measurement system that combines miniaturization and robust alignment. While the system’s ability to handle environmental changes has not yet been evaluated, the use of an optical fiber slot for light incidence is expected to enhance vibration resistance. Since the detector and light source are sufficiently capable of handling temperature changes, the system is also considered suitable for use in outpatient settings.

## Figures and Tables

**Figure 1 sensors-25-04368-f001:**
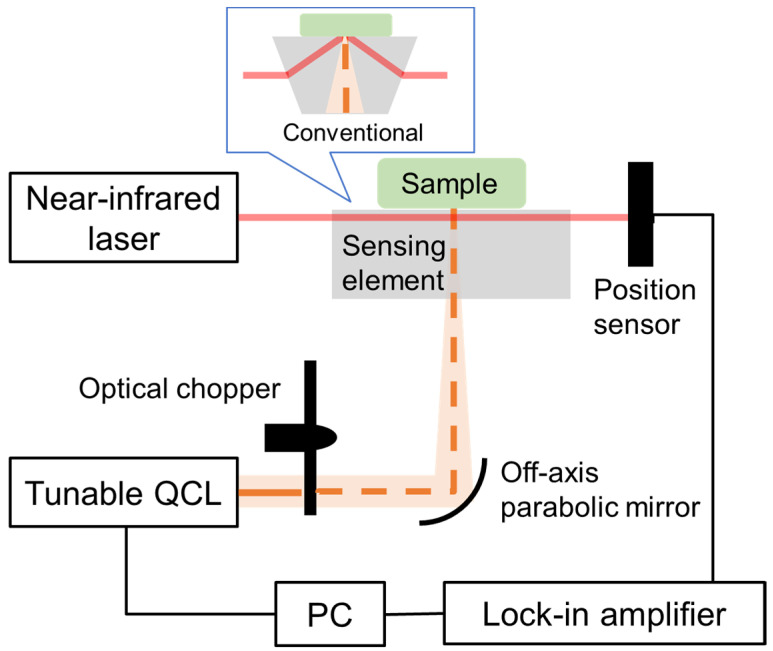
Schematic of the experimental setup.

**Figure 2 sensors-25-04368-f002:**
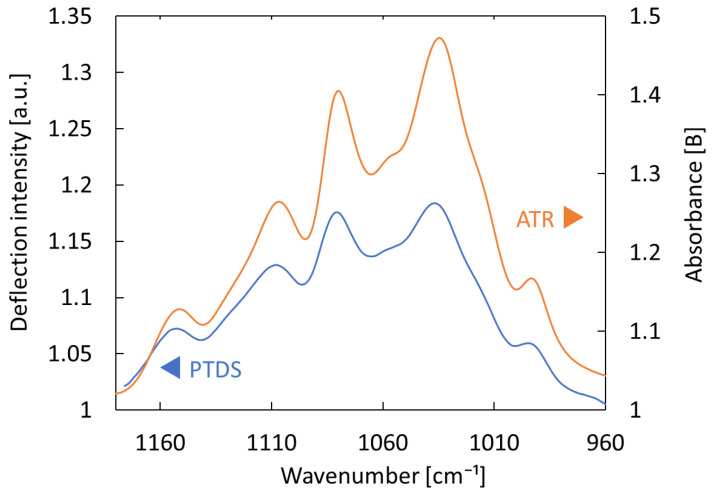
Comparison between absorption from photothermal deflection spectroscopy (PTDS) and attenuated total reflection (ATR) methods with 5% glucose solution.

**Figure 3 sensors-25-04368-f003:**
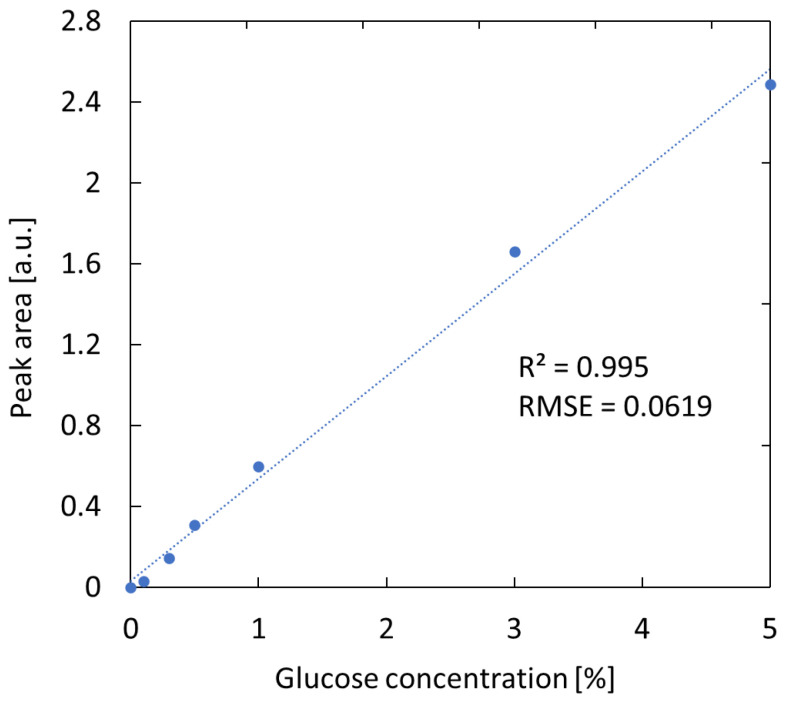
Relationship between the absorption peak area and the concentration of glucose solutions.

**Figure 4 sensors-25-04368-f004:**
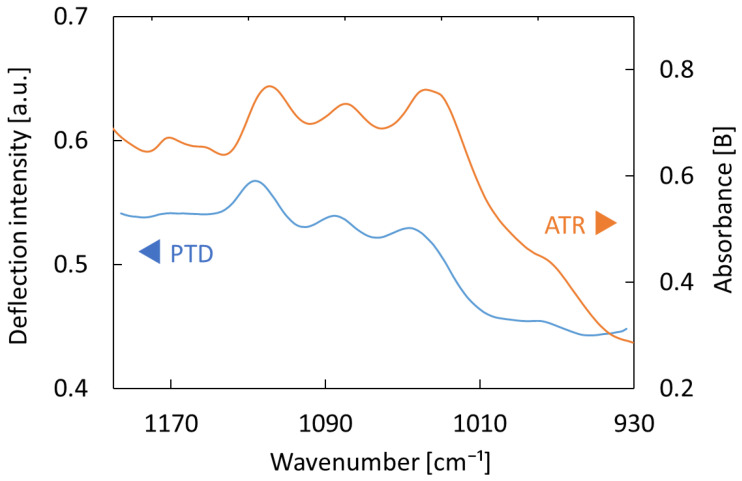
PTDS spectrum of a human wrist compared with an ATR spectrum.

**Figure 5 sensors-25-04368-f005:**
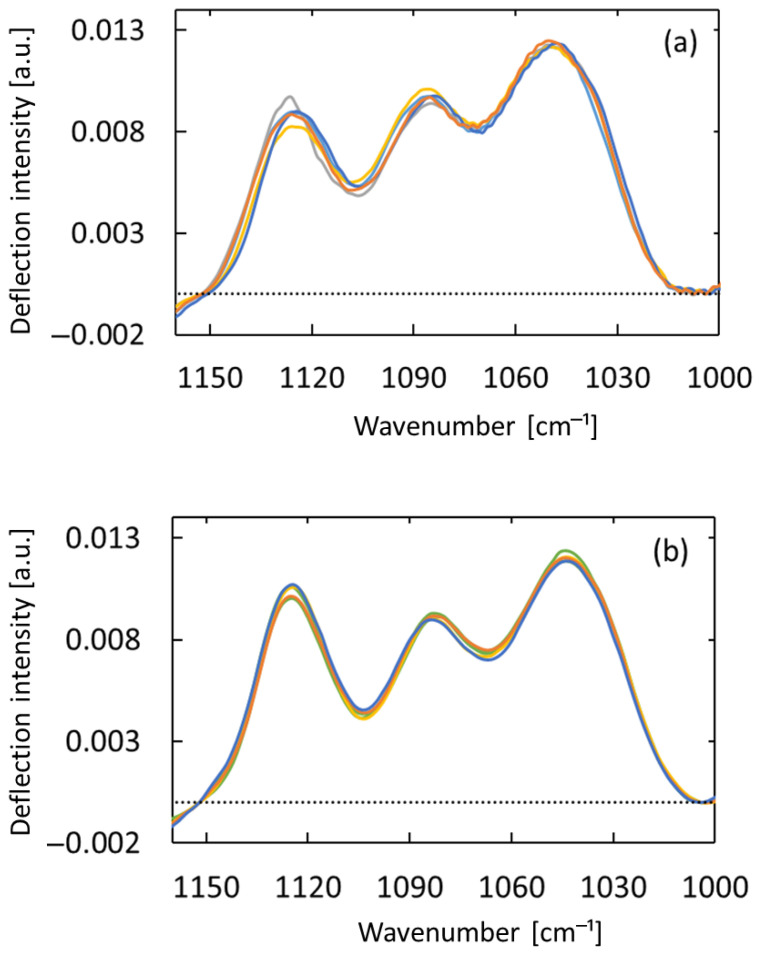
Continuous measurements obtained with continuous contact between the wrist and sensing element between measurements. Spectra were obtained with (**a**) the total reflection light path or (**b**) the horizontal light path.

**Figure 6 sensors-25-04368-f006:**
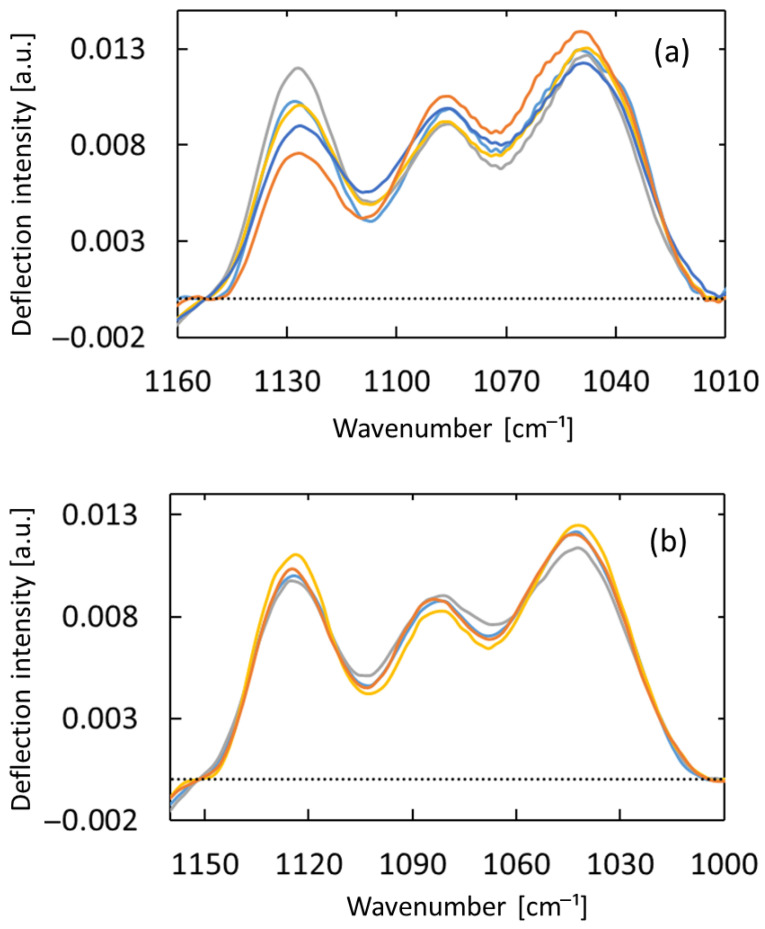
Measurements obtained with removal, rinsing, and repositioning of the sensing element between measurements. Spectra were obtained with (**a**) the total reflection light path or (**b**) the horizontal light path.

**Table 1 sensors-25-04368-t001:** Coefficients of variation for three absorption peaks calculated for each measurement.

	Continuous Contact Between Measurements		Repositioning Between Measurements	
Optical Path	Total Reflected	Horizontal	Total Reflected	Horizontal
1044 cm^−1^	2.01%	2.27%	3.57%	3.51%
1084 cm^−1^	1.80%	2.18%	5.67%	2.45%
1126 cm^−1^	0.95%	1.25%	16.47%	3.65%

## Data Availability

The spectral data used in this study are available from the corresponding author upon reasonable request.

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
