# Peer review of "Noninvasive Analysis of Biological Components Using Simplified Mid-Infrared Photothermal Deflection Spectroscopy"

_sensors, 2025, doi:10.3390/s25144368_

Round 1

Reviewer 1 Report

Comments and Suggestions for Authors

The paper presents a study on the non-invasive analysis of biological tissues using a simplified configuration of mid-infrared photothermal deflection spectroscopy (PTDS). The proposed system eliminates the complex alignment requirements of traditional PTDS, improving measurement reproducibility and allowing for a more practical path towards clinical implementation. The authors validate their system through comparative tests with conventional total internal reflection configurations, evaluating the reproducibility of spectra on aqueous glucose solutions and human wrist tissue. However, the manuscript lacks integration with physiological or hybrid detection systems, an absence of long-term usability tests, sensor lifespan analysis, or hardware implementation potential. Furthermore, the paper does not include references to AI-assisted spectral deconvolution or multivariate analysis methods. For such critical issues, the authors must respond to the following observations:

1) How can the proposed PTDS system be integrated into patient monitoring platforms that feature wearable, low-energy, and real-time sensors? I do not expect the authors to implement a new study from scratch; I only ask them to include in Section 4 Conclusions, when discussing future applications, and in the references, the present study doi: 10.3390/electronics13040790. This work shows how embedded systems with optimised optical components are already used in home care and could serve as a framework for the implementation of PTDS in wearable or bed contexts. The integration into embedded platforms for home monitoring can follow principles similar to those used in ventilator-based diagnostics.

2) What are the long-term durability and biocompatibility characteristics of the infrared-transparent glass used?

3) Could SAR or thermal measurements improve the interpretation of the PTDS signal in multilayer tissues or compensate for the low signal absorption?

4) Is the system adaptable for multimodal detection (for example, combining PTDS with sEMG or optical impedance mapping)?

5) What specific multivariate spectral analysis techniques are intended for the future classification of blood components?

6) Is it possible to miniaturise the current optical design while maintaining the robustness of the alignment?

7) How does the system respond to movement, temperature drift, or environmental changes typical of outpatient settings?

8) Can photothermal deflection be improved through bioimpedance-based tissue modelling to better target IR absorption sites?

9) All figures must present better quality.

Author Response

Comment 1

How can the proposed PTDS system be integrated into patient monitoring platforms that feature wearable, low-energy, and real-time sensors? I do not expect the authors to implement a new study from scratch; I only ask them to include in Section 4 Conclusions, when discussing future applications, and in the references, the present study doi: 10.3390/electronics13040790. This work shows how embedded systems with optimised optical components are already used in home care and could serve as a framework for the implementation of PTDS in wearable or bed contexts. The integration into embedded platforms for home monitoring can follow principles similar to those used in ventilator-based diagnostics.

Response 1

The PTDS system proposed in this study has the potential to be applied to wearable sensors for home healthcare by miniaturizing the system. The reference suggested by the reviewer is a pioneering model implementation of medical equipment and a similar integration approach may be possible when the system in this study is commercialized as a tabletop device. We have added the paper suggested by the reviewer to the references and incorporated the above comments into the revised manuscript in the Conclusion.

Comment 2

What are the long-term durability and biocompatibility characteristics of the infrared-transparent glass used?

Response 2

The stability of the infrared-transparent glass used in this paper is shown in the newly added reference 23. In addition, this glass does not use harmful substances such as arsenic and selenium in its materials, so there are no issues with biocompatibility. The above content has been added to the text (page 4).

Comment 3

Could SAR or thermal measurements improve the interpretation of the PTDS signal in multilayer tissues or compensate for the low signal absorption?

Response 3

Thermal measurements may be effective in interpreting PTDS signals caused by heat generated by light absorption. For PTDS targeting multilayer tissues, it may be possible to select the measurement depth by changing the modulation frequency, and this is currently under consideration. This information has been added to the text (page 6).

Comment 4

Is the system adaptable for multimodal detection (for example, combining PTDS with sEMG or optical impedance mapping?

Response 4

In particular, when combined with non-optical measurements such as sEMG, it may be possible to detect the effects of tissue structures that cannot be detected by light alone. PTDS does not interfere with electrical measurements, so it is technically feasible, and we will consider this in the future. We have added this information to the Conclusion.

Comment 5

What specific multivariate spectral analysis techniques are intended for the future classification of blood components?

Response 5

We have already begun introducing a method using partial least squares regression (PLSR), which has been reported in previous studies, and plan to report the results in the near future. In addition, machine learning methods such as random forests and neural network are also effective. We have added this information to the revised manuscript (page 6).

Comment 6

Is it possible to miniaturize the current optical design while maintaining the robustness of the alignment?

Response 6

The probe light input position is optimized by simulation calculation, and an optical fiber slot integrated with the glass sensing element makes it possible to input the probe light from an optical fiber equipped with a collimator without alignment, thereby enabling both miniaturization and robust alignment. We are currently designing this system because it enables both compactness and robustness of alignment. This information has been added to the text (page 10).

Comment 7

How does the system respond to movement, temperature drift, or environmental changes typical of outpatient settings?

Response 7

Currently, the system is built on a breadboard in a laboratory, and its ability to cope with environmental changes has not yet been investigated. However, as mentioned above, the use of optical fiber injection is expected to enhance resistance to vibration. In addition, since the detector and light source can sufficiently cope with temperature changes even at present, it is expected that the system will be able to cope with ambulatory care environments in the near future. This information has been added to the Conclusion.

Comment 8

Can photothermal deflection be improved through bioimpedance-based tissue modelling to better target IR absorption sites?

Response 8

If the amount of water and fat in the tissue can be determined by bioimpedance-based tissue modeling, the physical simulation of PTDS that we are currently studying may be more accurate and improve the accuracy of its measurement. We will consider this based on the reviewer’s suggestions. This information has been added to the text (page 9).

Comment 9

All figures must present better quality.

Response 9

All figures have been revised with higher resolution for better readability.

Reviewer 2 Report

Comments and Suggestions for Authors

This paper presents a photothermal deflection spectroscopy (PTDS) to non-invasively measure biological tissues. While the previous PTDS system relies on total internal reflection, the proposed method employs a horizontal optical path, offering simplified optical alignment and improved measurement reproducibility.

The authors validated their system using both in vitro (glucose) and in vivo (human wrist) against the conventional ATR based techniques, showing a good agreement in absorption spectra. Also, they demonstrated higher reproducibility in human study through repeated measurements, reported as CV. While technical improvement is notable, several things must be addressed before publication.

The comparison between two methods is largely qualitative and limited to glucose. For example, Figure 3 lacks statistical metrics to support linearity. Broader testing on other biologically relevant analytes would strengthen the manuscript. The claim of detecting 0.3% glucose is not well supported by the results. The authors need clearer explanation on how the detection threshold was determined and whether this corresponds to physiologically relevant blood glucose concentrations.  

Author Response

Comment 1

Figure 3 lacks statistical metrics to support linearity.

Response 1

We appreciate very much the suggestion. To evaluate the linearity in Fig. 3, we added the coefficient of determination R2 and the root mean square error (RMSE), a measure of error relative to the approximate straight line.

Comment 2

Broader testing on other biologically relevant analytes would strengthen the manuscript.

Response 2

In this paper, glucose was selected as a model substance to demonstrate the feasibility of the system, but in principle, various other blood components can also be detected. Our group has successfully analyzed blood cholesterol using the mid-infrared ATR method, and similar analysis is possible for PTDS. This is described in the text (page 7) along with a new reference 25.

Comment 3

The claim of detecting 0.3% glucose is not well supported by the results. The authors need clearer explanation on how the detection threshold was determined and whether this corresponds to physiologically relevant blood glucose concentrations. 

Response 3

We appreciate the precise comments. Indeed, as pointed out by the reviewer, at present the claim of 0.3% glucose detection was not adequately supported by the results available for publication. Since the relationship between blood glucose concentration and optical absorption detected from the body surface is still under investigation, we have removed that statement from the paper and added the following statement: "A glucose concentration of 0.3% corresponds to a human blood glucose level of 300 mg/dL, and the sensitivity is still insufficient to detect fasting blood glucose levels of approximately 80–200 mg/dL in healthy subjects. We are currently studying the optimization of the probe light incident position by simulation calculation to improve the sensitivity.

Reviewer 3 Report

Comments and Suggestions for Authors

In this work, the authors compare different system configurations in mid-infrared photothermal deflection spectroscopy (PTDS). While the work seems interesting, I would like to invite the authors to revise the articles addressing questions and comments shown below.

The references are relatively outdated. Could the authors cite more relevant articles that were published recently (e.g., published in 2022 to 2025)?

Could the authors compare the optical properties of crystalline ZnS and the sensing element introduced in the work? How about the costs of the two materials?

Could the authors add an explanation on why heat-sensing medium is or should be used in mid-infrared PTDS?

No information on sample preparation is provided.

The unit of wavelength is not cm^-1. It seems the authors are confused about wavelength and wavenumber.

The authors mentioned an issue at lines 145 and 146. I believe more information should be provided on how the authors addressed the issue.

I don’t understand what the authors mean by ‘details’ on line 148. I am not sure whether it is OK not to unveil the method used in the work.

Please provide relevant references for the descriptions between lines 160 and 166. Also, please provide reference(s) for line 169.

What do authors mean by ‘these absorption levels’ on line 166?

Could the authors provide more descriptions of ‘each experiment’ on line 176?

Could the authors provide more descriptions of ‘continuous’ on line 183?

What are the straight lines in Figures 5 and 6?

Why do we see very different spectra when comparing Figure 4, and Figures 5 and 6, even considering that Figures 5 and 6 show normalized intensity?

What do authors mean by ‘optical path’ in Table 1?

I do not understand what the authors mean by ‘0.3%’ on line 214.

The manuscript should have more detailed discussion comparing the work with previous works.

Author Response

Comment 1

The references are relatively outdated. Could the authors cite more relevant articles that were published recently (e.g., published in 2022 to 2025)?

Response 1

We appreciate the suggestion. We have added more recently published and more relevant papers as the reviewer indicated, Refs. 16, 24, 25.

Comment 2

Could the authors compare the optical properties of crystalline ZnS and the sensing element introduced in the work? How about the costs of the two materials?

Response 2

The refractive index and transmission wavelength range of the infrared-transparent glass used in this study were compared with those of ZnS, which has been used conventionally in the revised text. It was also noted that although the cost of the materials is similar, glass is suitable for future mass production because it can be molded. (page 4)

Comment 3

Could the authors add an explanation on why heat-sensing medium is or should be used in mid-infrared PTDS?

Response 3

In conventional solid material evaluation systems using PTDS with visible light or near-infrared light, changes in refractive index due to temperature increases in the air were utilized. However, in biomedical application systems using mid-infrared light, it is necessary to use a solid heat-sensing medium to securely fix soft samples such as skin. Additionally, the infrared-transparent glass used in this study has a thermal optical coefficient dn/dT of 3.0×10⁻⁵, which is significantly higher than the

-9.7×10⁻⁷ of air, potentially enabling much higher sensitivity. This information is described in the text (page 4).

Comment 4

No information on sample preparation is provided.

Response 4

Detailed information on sample preparation has been added (page 4).

Comment 5

The unit of wavelength is not cm⁻¹. It seems the authors are confused about wavelength and wavenumber.

Response 5

As the reviewer pointed out, the text contained a mixture of wavelength and wavenumber notation. In mid-infrared spectroscopy, wavenumber is the standard unit, so we have revised the text in the relevant section to use wavenumber to avoid misunderstanding.

Comment 6

The authors mentioned an issue at lines 145 and 146. I believe more information should be provided on how the authors addressed the issue.

Response 6

We appreciate the precise comments. Indeed, as pointed out by the reviewer, at present the claim of 0.3% glucose detection was not adequately supported by the results available for publication. Since the relationship between blood glucose concentration and optical absorption detected from the body surface is still under investigation, we have removed that statement from the paper and added the following statement: "A glucose concentration of 0.3% corresponds to a human blood glucose level of 300 mg/dL, and the sensitivity is still insufficient to detect fasting blood glucose levels of approximately 80–200 mg/dL in healthy subjects. We are currently studying the optimization of the probe light incident position by simulation calculation to improve the sensitivity.

Comment 7

I don’t understand what the authors mean by ‘details’ on line 148. I am not sure whether it is OK not to unveil the method used in the work.

Response 7

As mentioned above, we are unable to provide detailed information at this stage, so we have deleted this description from the main text and added the comment above.

Comment 8

Please provide relevant references for the descriptions between lines 160 and 166. Also, please provide reference(s) for line 169.

Response 8

We added a reference related to the descriptions in lines 160 to 166 (Ref. 18) and a reference related to line 169 (Ref. 24).

Comment 9

What do authors mean by ‘these absorption levels’ on line 166?

Response 9

“These absorption levels” refer to absorption in the interstitial fluid beneath the stratum corneum and are much lower than the absorption levels of the stratum corneum. We revised the text to make it clearer.

Comment 10

Could the authors provide more descriptions of ‘each experiment’ on line 176?

Response 10

“Each measurement took two min.” means that it took two minutes to obtain one spectrum. In the continuous measurement, the spectrum was measured five times in a row while the skin remained in contact with the sensing element. In the repositioning measurement, the skin was removed from the sensing element after each spectrum measurement. We revised the text to make it clearer.

Comment 11

Could the authors provide more descriptions of ‘continuous’ on line 183?

Response 11

As mentioned above, in the continuous measurement, the spectrum was measured five times in a row while the skin remained in contact with the sensing element. We revised the text to make it clearer.

Comment 12

What are the straight lines in Figures 5 and 6?

Response 12

Since baseline fluctuations occur in the measured PTDS spectrum, the baseline has been corrected in the 1005–1152 cm⁻¹ range to compensate for this. The baselines are shown in Figs. 5 and 6. The spectrum has been normalized by the area between the line and the spectrum. We revised the text to make it clearer.

Comment 13

Why do we see very different spectra when comparing Figure 4, and Figures 5 and 6, even considering that Figures 5 and 6 show normalized intensity?

Response 13

The spectra in Figures 5 and 6 are raw spectra (Fig. 4) rotated based on the above baseline.

Comment 14

What do authors mean by ‘optical path’ in Table 1?

Response 14

“Optical path” refers to the incident light path of the probe light to the sensing prism. “Total reflected” refers to measurements in a total reflection light path system, and “Horizontal” refers to measurements in the horizontal light path system proposed in this study.

Comment 15

I do not understand what the authors mean by ‘0.3%’ on line 214.

Response 15

“0.3%” refers to the concentration of glucose solution, which is equivalent to a human blood sugar level of 300 mg/dL.

Comment 16

The manuscript should have more detailed discussion comparing the work with previous works.

Response 16

We added a comparison with previous studies to the conclusion.

Round 2

Reviewer 1 Report

Comments and Suggestions for Authors

I thank the authors for their replies to the comments. I have no further comments to ask. 

Reviewer 2 Report

Comments and Suggestions for Authors

The author has adequately addressed my main three comments by adding a statistical metric, removing the unsupported sensitivity and mentioning more prior works to support their applicability. 

Reviewer 3 Report

Comments and Suggestions for Authors

Generally, the authors revised properly based on the provided comments.

I recommend the acceptance of the manuscript for publication in the journal.

Meanwhile, please add partial least squares regression (PLSR) in the list of abbreviations.